# Optimization of the Growth Process of Double Perovskite Pr_2−δ_Ni_1−x_Mn_1+x_O_6−y_ Epitaxial Thin Films by RF Sputtering

**DOI:** 10.3390/ma15145046

**Published:** 2022-07-20

**Authors:** Mónica Bernal-Salamanca, Lluis Balcells, Zorica Konstantinović, Alberto Pomar, Benjamín Martínez, Carlos Frontera

**Affiliations:** 1Institut de Ciència de Materials de Barcelona, Consejo Superior de Investigaciones Científicas (ICMAB-CSIC), Campus UAB, 08193 Bellaterra, Spain; balcells@icmab.es (L.B.); apomar@icmab.es (A.P.); bejamin.martinez@icmab.es (B.M.); frontera@icmab.es (C.F.); 2Center for Solid State Physics and New Materials, Institute of Physics Belgrade, University of Belgrade, Pregrevica 68, 6080 Belgrade, Serbia; zorica@ipb.ac.rs

**Keywords:** ferromagnetic insulator, double perovskite, sputtering thin film growth

## Abstract

Epitaxial thin films of Pr_2−δ_Ni_1−x_Mn_1+x_O_6−y_ (PNMO) double perovskite were grown on (001)-oriented SrTiO_3_ substrates by RF magnetron sputtering. The influence of the growth parameters (oxygen pressure, substrate temperature, and annealing treatments) on the structural, magnetic and transport properties, and stoichiometry of the films was thoroughly investigated. It is found that high-quality epitaxial, insulating, and ferromagnetic PNMO thin films can only be obtained in a narrow deposition parameter window. It is shown that a careful selection of the growth conditions allows for obtaining a high degree of Ni/Mn cation ordering, which is reflected in the values of the Curie temperature, T_C_, and saturation magnetization, M_S_, which are very close to those of bulk material.

## 1. Introduction

Spintronics has been pointed out as a potential alternative technology for the development of multifunctional electronics combining data storage and transmission, as well as logical operations, with improved energy efficiency. The generation and control of pure spin currents [1] or highly polarized electron currents [2] is at the cornerstone of spintronics. One of the most common ways for generating a highly polarized electron current is the spin filtering through the ferromagnetic–insulator–ferromagnetic tunnel barriers [3]. In addition, the same can be achieved by using a single element, a ferromagnetic-insulating (FM-I) barrier [4,5]. However, FM-I materials are very scarce. In this regard, double perovskite (DP) oxides of the R_2_NiMnO_6_ family (RNMO, where R is a rare earth element) have received considerable attention for presenting this behavior. Their multifunctional physical properties make them potential candidates for technological applications beyond spintronic devices, such as magnetodielectric capacitors and multiple state logic devices [6,7,8,9,10]. In this family, La_2_NiMnO_6_ (LNMO; R = La) has attracted much more attention for its near-room-temperature ferromagnetic transition (T_C_ ≈ 280 K), insulating characteristics, magneto-dielectric effect, and spin–phonon coupling [7,11,12]. The ferromagnetic (FM) ordering in RNMO systems is explained in terms of the superexchange interaction between Ni^2+^ $\left(d^{8},t_{2g}^{6}e_{g}^{2},\;S=1\right)$ and Mn^4+^ $\left(d^{3},t_{2g}^{3}e_{g}^{0},S=3/2\right)$, which is FM in 180° geometry according to the Goodenough–Kanamori rules [7,13,14,15]. For this reason, the magnetic properties of these materials are found to be very sensitive to the Ni/Mn cationic ordering inside of the DP structure.

Previous studies on RNMO systems have demonstrated that the substitution of La^3+^ by another rare earth element, R^3+^, with a smaller ionic radius induces significant modifications in the Ni–O–Mn bond angle directly involved in the superexchange mechanism [15,16,17]. Consequently, compared to that of LNMO, the magnetic transition temperature, T_C_, decreases monotonously with decreasing the $r_{R^{3+}}$ radius and the concomitant increasing of the octahedral tilting [15,17]. In this regard, the structural, magnetic, and dielectric properties are strongly dependent on the size of the rare earth element. Over the years, the synthesis and physical properties of the RNMO (especially for R = La) compounds in bulk form [18,19] have been widely studied in order to gain a better understanding of the nature of the magnetic exchange interactions [18,19,20]. However, despite the great interest, and in contrast to the LNMO thin films, the investigations of the other members of the RNMO family such as Pr_2_NiMnO_6_ (PNMO) in thin film form are very scarce. In bulk, PNMO has a single ferromagnetic-to-paramagnetic transition at 228 K and a monoclinic symmetry with the space group *P* 2_1_/*n* [21]. In this context, studies have concluded that ordered RNMO double perovskite thin films can only be stabilized within a very narrow growth window [22,23]. Outside this window, studies on thin films have evidenced that there is either a short-range-ordered phase or the coexistence of multiple phases, such as the polycrystalline bulk samples in thin films of LNMO [7,24]. Physical deposition methods, such as pulsed laser deposition (PLD), have been amongst the few techniques used to prepare RNMO thin films. Particularly, so far, there are few reports on the synthesis routes and growth conditions (such as pressure, temperature, annealing atmosphere/temperature, and deposition time) of PNMO thin films, which have a strong influence on the crystallographic structure and ferromagnetic properties of the films [15,16,25,26,27,28].

In this work, we report on the growth optimization and characterization of PNMO thin films deposited on top of (001)-oriented SrTiO_3_ (STO) substrates by the RF magnetron sputtering technique. The influence of the growth parameters (i.e., oxygen pressure, substrate temperature, and annealing treatments) on the structural and magnetic properties, as well as the stoichiometry of the films, is thoroughly investigated. After the optimization of the growth conditions, we demonstrate that high-quality epitaxial, insulating, and ferromagnetic PNMO thin films with good B-site cationic ordering (i.e., T_C_ and M_S_ very close to the bulk value) can be obtained by the RF sputtering technique despite a certain degree of nonstoichiometry.

## 2. Experimental Details

The PNMO thin films were deposited on (001)-oriented STO substrates by the RF magnetron sputtering technique by using a stoichiometric Pr_2_NiMnO_6_ target prepared by the solid-state reaction method. Before the deposition, the substrates were cleaned in an ultrasonic bath with Milli-Q water and then annealed at 1000 °C in air for 2 h to obtain a clean and smooth TiO_2_-terminated surface [29]. Stoichiometric target (φ = 1.3”) was synthesized by mixing the primary oxide powders (99.99% pure), i.e., La_2_O_3_, Pr_2_O_3_, MnO_2_, and NiO. At the end, the pressed pellet was thermally treated up to 1400 °C for 48 h in air, as this process ensures to achieve a homogeneous and dense target. The X-ray powder diffraction pattern analysis of the final target of Pr_2_NiMnO_6_ with no additional spurious phase present is shown in Appendix A. The XRD peaks were indexed on the basis of monoclinic *P* 2_1_/*n* (single cell) according to the Rietveld refinement. The average of the lattice parameters of the pseudocubic phase of Pr_2_NiMnO_6_ was 3.871 Å.

Films were grown at different background oxygen pressures (100–750 mTorr) and growth temperatures ranging from 200 to 900 °C. After thin film growth, the films were annealed in situ at the same growth temperature for 1h under 420 Torr O_2_ and then slowly cooled down to room temperature at 10 °C/min. All PNMO films were prepared at a fixed RF power (40 W) and a fixed target-to-substrate distance (5 cm).

The surface morphology of the samples was characterized by scanning electron microscopy (SEM, QUANTA 200 FEG-ESEM, FEI Company, Hillsboro, OR, USA) and atomic force microscopy (AFM, MFP-3D AFM, Asylum Research, Santa Barbara, CA, USA) in tapping mode. Structural characterization of the samples, crystallinity quality, and thickness was studied by X-ray diffraction (XRD) and X-ray reflectivity (XRR) using the Bruker D8-Advance (Berlin, Germany) and Siemens D5000 diffractometers (Cu-Kα_1_ and Cu-Kα_1,2_ radiation, respectively).

The chemical composition and stoichiometry ratio of the PNMO thin films were determined by wavelength dispersive spectrometer (WDS) electron probe microanalysis (EPMA) using a (CAMECA, Gennevilliers, France) SX-50 electron microprobe equipped with four wavelength-dispersive X-ray spectrometers. The electrical transport properties were measured in a Physical Properties Measurement System (PPMS, Quantum Design, San Diego, CA, USA) by the four-probe technique. Magnetization measurements were done CA a superconducting quantum interferometer device (SQUID, Quantum Design).

## 3. Results and Discussion

### 3.1. Oxygen Pressure Deposition

First, the optimal deposition conditions for growing the ordered double perovskite PNMO thin films on SrTiO_3_ substrates were examined. We started the optimization of the growth parameters using high growth temperatures (850 °C), following previous studies [26]. Under these conditions, a series of PNMO thin films were grown for 1 h under different oxygen pressures (ranging from 100 to 750 mTorr) by keeping fixed the growth temperature (T = 850 °C). Appendix A shows the XRR measurements that were carried out to determine the thickness of the samples. Thicknesses were obtained from the positions of the minima in the XRR curve. In Appendix A, one can appreciate that when the oxygen pressure increases from 100 to 750 mTorr, the thickness decreases monotonously from 95 nm to 28.5 nm. Accordingly, the growth rate decreases from 1.6 to 0.5 nm/min, respectively. Normally, by increasing the oxygen pressure in the chamber, the plasma becomes broader, and the mean free path of the atoms decreases, thus decreasing the quantity of the arriving adatoms and the growth rate.

The crystallinity quality of the films has been evaluated by X-ray diffraction. Figure 1a shows θ/2θ scans of the PNMO thin films at a fixed deposition temperature of 850 °C under various oxygen pressures. On all the XRD patterns, the (00*l*) reflections of the PNMO phase (indexed in the pseudocubic notation), which take place at 2θ values slightly higher than the ones of SrTiO_3_, can be clearly observed (see dashed vertical red lines), thus indicating a good crystalline quality of the films. For closer inspection, Figure 1b shows the (002) reflection of both the STO substrate and the PNMO film for each sample. One can observe that the position of the (002) peak from the PNMO on the STO occurs at higher 2θ angles than the bulk PNMO target value of 3.871 Å located at 2θ = 46.9° (see dashed vertical red line in Figure 1b). This indicates a shrinking of the out-of-plane *c* lattice parameter, in agreement with the lattice mismatch imposing a tensile in-plane strain. In Figure 1c, the estimated out-of-plane *c* lattice parameter is plotted. At 100 mTorr O_2_, the PNMO (002) reflection is located at 2θ ≈ 47.17°, corresponding to an out-of-plane cell parameter of 3.864 Å. By increasing the oxygen pressure from 100 to 300 mTorr, the PNMO (002) peak shows a shift towards the slightly higher 2θ angles (see arrow) if compared with its bulk position. This displacement is associated with a decrease of the out-of-plane *c* lattice parameter (see Figure 1c). At intermediate oxygen pressures (350 and 400 mTorr), a shift of the PNMO peak towards lower angles can be seen (see arrow), indicating an expansion of the out-of-plane *c* lattice parameter of the PNMO film, approaching the bulk PNMO value of 3.871 Å (see Figure 1c). With the further increasing of the oxygen pressure (above 400 mTorr), the out-of-plane *c* lattice parameter decreases again. The latter could be associated with a variation in the concentration of the defects and oxygen vacancies into the structure under different oxygen pressures.

Finally, in addition to the PNMO reflections, the XRD patterns show additional reflections corresponding to the parasitic phases. In all the XRD patterns of the samples grown at very high and low oxygen pressures, the (002) reflection of the NiO phase can be clearly observed at 2θ ≈ 43.4° (see the asterisk and dashed blue line in Figure 1a,b). At the intermediate oxygen pressure (from 300 to 400 mTorr), a slight shift of this peak towards greater angles can be observed around 2θ ≈ 43.7°. This slight shift in the NiO peak could be attributed, at first glance, to a change in the oxygen content of the nickel oxide (NiO_x_). On the other hand, the reason for the presence of the NiO phase in the PNMO samples (grown/annealed at 850 °C) is not well understood, although this phase segregation may be of the same origin as seen in the LNMO thin films, as previously reported [30]. Moreover, it could be a consequence of a certain degree of Pr deficiency, as revealed by the electron probe microanalysis (EPMA) measurements (see below). This Pr deficiency may indeed induce the formation of secondary phases in the R_2_NiMnO_6_ double perovskite, as reported by Dass et al. [19]. Similarly, a secondary phase located at 43.24°, identified as NiO, was also observed in the XRD pattern of the Nd_2_NiMnO_6_ samples [31]. Therefore, according to the XRD data, the PNMO films grown at 850 °C under different oxygen pressures have a good crystalline quality. In addition, although the secondary phases are formed (NiO) together with the PNMO phase, the films grow epitaxial on the (001) STO substrates.

The surface morphology of the PNMO films was examined by AFM and SEM measurements (see Figure 2a,b and Appendix A). All the AFM images (left-hand images) show a granular morphology (no step–terrace morphology detected). Furthermore, in Figure 2c, one can appreciate that the PNMO films deposited under low oxygen pressures (ranging from 100 to 300 mTorr) have high root mean square (rms) roughness values (rms > 1.5 nm). By increasing the oxygen pressure from 350 to 750 mTorr, the film morphologies become flatter and smoother with low roughness values (rms ≈ 0.5 nm). At the same time, the presence of very fine grains distributed all over the film surface can be observed. The lower roughness could be related to the fact that the films grown under high oxygen pressure are thinner (see Appendix A). However, the decrease of the film thickness with oxygen pressure is monotonous, while the roughness decreases abruptly above 350 mTorr, indicating that the growth process at higher pressure favors a flatter surface.

Additionally, the effect of the oxygen pressure on the film morphology was also explored by the SEM measurements. Figure 2a,b and Appendix A show the SEM images (right-hand images) of the corresponding AFM ones. Films deposited at low oxygen pressures (PO_2_ < 350 mTorr) show a rather flat morphology covered with small grains distributed homogeneously over the film surface. The latter corroborates the morphology observed by AFM and the increase of the roughness of the films. On the other hand, at high oxygen pressures (PO_2_ ≥ 350 mTorr), the SEM images reveal a flat and smooth morphology, although some small grains randomly distributed over the film surface can be observed. As mentioned before, high pressure favors a smoother surface. The presence of these small grains at the film surface may be due to the presence of the NiO secondary phase, as previously detected by the XRD patterns (see Figure 1a,b). On the other hand, the existence of a nanostructured surface was also detected in the LNMO systems, and it was attributed to the formation of NiO_x_ aggregates after high annealing temperatures (800–900 °C) [30].

### 3.2. Deposition Temperature

According to the results on the PNMO films grown at different oxygen pressures, good ferromagnetic properties were obtained for the samples deposited at intermediate oxygen pressures ranging from 350 to 400 mTorr O_2_ (vide ultra). Therefore, in order to further optimize the deposition conditions, the PNMO thin films on the STO (001) substrates were grown at different substrate temperatures (ranging from 200 °C to 900 °C) by keeping a fixed oxygen pressure of 350 mTorr for 1 h. Appendix A shows a typical XRR curve of a PNMO film deposited at high temperature (T = 700 °C). The thickness dependence as a function of the growth temperature is shown Appendix A. It can be observed that the film thickness remains almost constant above 600 °C. The thickness values found via the XRR analysis for all the PNMO samples grown at different temperatures are very similar (~48 nm), with a growth rate of about 0.8 nm/min. In a particular case, the sample deposited at a substrate temperature of 200 °C was thicker (~54 nm).

Figure 3a shows the XRD θ/2θ scans of the ~48 nm-thick PNMO films grown under 350 mTorr O_2_ at different temperatures. In addition, in all the XRD patterns, only the (00*l*) reflections of the PNMO phase were clearly observed in the samples grown at temperatures above 600 °C (see the dashed vertical red lines), indicating that high growth temperatures favor a good crystalline quality. For more details, Figure 3b shows the (002) reflection of both the STO substrate and the PNMO film. As the growth temperature increases, it can be observed that the position of the PNMO (002) peak shows a shift towards larger angles (see arrow) if compared with the position of the PNMO bulk value of 3.871 Å located at 2θ= 46.9° (see the dashed vertical red line in Figure 3b). This shift indicates a progressive shrinking of the out-of-plane *c* lattice parameter (see Figure 3c) due to the tensile in-plane strain imposed by the substrate. The *c* lattice parameter of the films decreases from 3.866 Å to 3.851 Å as the growth temperature increases from 600 °C to 900 °C, as shown in Figure 3c.

In addition to the PNMO reflections, the XRD patterns show the existence of a secondary NiO phase (see the asterisk and dashed blue line in Figure 3a,b), as previously detected in the samples deposited at different pressures (Figure 1). Nevertheless, for the films grown/annealed in the 600–700 °C range, the (002) NiO reflection has a very low intensity. As the growth temperature increases, it can be observed that the NiO peak is slightly more intense and presents a slight shift towards lower angles (2θ ≈ 43.8° to 43.6° for 800–900 °C, respectively). As discussed before, this phase segregation may be of the same origin as seen in the LNMO thin films [30]. Therefore, according to the XRD patterns, the PNMO films deposited under 350 mTorr O_2_ at different temperatures grow epitaxially on the STO substrates, and with a good crystalline quality.

In order to determine the values of the in-plane (*a*) and out-of-plane (*c*) lattice parameters, a reciprocal space map (RSM) around the (−103) reflection was performed on an illustrative PNMO film with optimized growth conditions (grown/annealed at 800 °C under 350 mTorr O_2_). The RSM (Figure 4) shows that the peak position q_x_ of the (−103) PNMO film reflection is slightly shifted towards a larger (absolute) value of q_x_ from that of the STO substrate. The latter is indicating that the in-plane (*a*) cell parameter of the PNMO film is slightly smaller than the STO one (a_STO_ = 3.905 Å). This contrasts with the PNMO films grown on top of (LaAlO_3_)_0.3_(Sr_2_AlTaO_6_)_0.7_ (LSAT) [27], although this can be attributed to the smaller lattice mismatch introduced by LSAT (a_LSAT_ = 3.869Å). Therefore, from the position of the (−103) PNMO film reflection, the in-plane lattice parameter (in pseudocubic notation) is found to be *a* = 3.877Å, which is closer to the lattice parameter of pseudocubic bulk PNMO (*a*_bulk_ = 3.871 Å), indicating a (partial) in-plane relaxation of the PNMO film. In addition, the out-of-plane lattice parameter of the PNMO film (*c*= 3.848 Å), determined from the q_z_ value of the (−103) PNMO reflection, is smaller than that of the STO, in agreement with the θ/2θ scans shown in Figure 3.

The AFM and SEM images of the PNMO films for the different growth temperatures are shown in Figure 5a,b and Appendix A. It can be observed that for the PNMO films grown at low temperatures, from 200 °C to 700 °C, the AFM and SEM images (left-hand images) show a flat surface without the presence of grains. Moreover, the roughness values of the films obtained from the AFM data are below 0.4 nm (see Figure 5c), indicating that the surface roughness is similar to that of the steps–terraces morphology of the underlying substrates. At higher growth temperatures (T ≥ 800 °C), the films show a granular morphology over the entire surface (see Appendix A). In particular, the sample grown at 900 °C has a surface morphology with a higher grain density (according to the SEM image in Appendix A), which is also reflected in an increase of the roughness (rms ≈ 0.7 nm see Figure 5c). Nevertheless, a low roughness value (rms < 1 nm) can be appreciated in all the PNMO films grown at 350 mTorr O_2_. In addition, the XRD patterns showed the existence of a secondary NiO phase (see Figure 3a,b). The NiO peak becomes more intense at high growth/annealing temperatures (800–900 °C), which could be related to the formation of NiO_x_ segregates at the film surface at high annealing temperatures (800–900 °C), as observed previously in the LNMO films [30]. For the films grown/annealed in the 200–700 °C range, the secondary NiO phase was not clearly detected by XRD, as corroborated by the AFM and SEM images.

### 3.3. Chemical Composition

The chemical composition and the stoichiometry ratio of the PNMO thin films grown on the STO substrates under different oxygen pressures and temperatures have been analyzed by EPMA. Measurements were done at 10 different points of each sample in order to estimate the error bars of the measurement. Figure 6a,b show the stoichiometry ratio obtained.

In Figure 6a, it can be appreciated that the stoichiometry ratio of the PNMO films (grown at 850 °C) shows a strong dependence on the oxygen pressure PO_2_. Samples grown at low oxygen pressures (ranging from 100 to 300 mTorr) present a Pr/(Ni+Mn) atomic slightly above 1 (see black symbols), while the Ni/Mn ratio is below 1 (see red symbols). One can conclude that these samples are Ni-deficient. Contrarily, in samples grown at high oxygen pressures (PO_2_ ≥ 350 mTorr), the Pr:(Ni+Mn) ratio is clearly below 1, while the Ni/Mn ratio is close to 1, indicating that the films have a certain degree of Pr deficiency. On the other hand, the error in determining the O content is larger than for the other elements due to the presence of oxygen in both the film and substrate. In this case, the oxygen concentration is not computed due to the presence of a large O background signal from the substrate, which is difficult to separate from the one coming from the film. Therefore, stoichiometry is not considered for the oxygen and the composition of the samples that are expressed as Pr_2−δ_Ni_1−x_Mn_1+x_O_6−y_ when Pr/(Ni+Mn) ≤ 1, and Pr_2_Ni_1−x_Mn_1−z_O_6−y_ when Pr/(Ni+Mn) ≥ 1, as the presence of oxygen vacancies (y > 0) cannot be ruled out. In this regard, the samples grown at 100 mTorr and 200 mTorr O_2_ show a Ni/Mn ratio of about 0.58 (Pr_2.0_Ni_0.7_Mn_1.2_O_6−y_) and 0.62 (Pr_2.0_Ni_0.7_Mn_1.1_O_6−y_), respectively, i.e., with Ni deficiency which thus affects the ferromagnetic properties (as discussed in the next subsection). At high oxygen pressures (PO_2_ ≥ 350 mTorr), the Pr:(Ni+Mn) ratio is below 1: films grown at 350 mTorr and 400 mTorr O_2_ show a Pr:(Ni+Mn) ratio of ~0.83 (Pr_1.7_Ni_0.9_Mn_1.1_O_6−y_) and ~0.73 (Pr_1.5_Ni_1.0_Mn_1.0_O_6−y_), respectively. The slight deficiency of Pr in the films when the oxygen pressure increases could be attributed to the fact of a greater scattering of plasma species, being that this scattering more notable for the lighter elements. Therefore, a deficient flow of adatoms on the film surface could render the formation of nonstoichiometric Pr/(Ni+Mn) < 1.

Finally, Figure 6b shows the atomic ratio of the PNMO films deposited at different growth temperatures for PO_2_ = 350 mTorr. It is found that all samples present a Pr/(Ni + Mn) atomic ratio clearly below 1, while Ni/Mn is near 1, indicating that the films show a Pr deficiency. At the same time, it can be also observed that by increasing the growth temperature (T ≥ 700 °C), the amount of Pr slightly increases but is still below the expected stoichiometry. For instance, the film grown/annealed at 800 °C shows Pr:(Ni + Mn) ≈ 0.86 and Ni/Mn ≈ 0.86 (Pr_1.7_Ni_0.9_Mn_1.1_O_6−y_).

### 3.4. Magnetic and Transport Properties

The magnetization as a function of temperature was measured by applying a 5 kOe magnetic field in the plane of the samples after a field cooling process (at the same field and orientation). The resulting curves are shown in Figure 7a,d. Curie temperatures were estimated from the inflection point of the M(T) curves and are plotted in Figure 7b,e. As can be seen, a quite significant magnetic moment appears in all films grown at 850 °C (Figure 7a), although the T_C_ (Figure 7b) presents a relevant dependence on the oxygen pressure between 100 mTorr and 350 mTorr. In this region, a fast increase of the Curie point from 118 K to about 200 K takes place. At 350 mTorr (and up to 750 mTorr), T_C_ reaches a plateau and remains almost constant around 200 K. In addition, in this growth pressure range, the M(T) curves do not show signs of any significant secondary magnetic transition at low temperature (in contrast with previously reported films grown by PLD [26,27]). These results clearly demonstrate that the ordering of Ni^2+^ and Mn^4+^ ions is strongly affected by the oxygen pressure during deposition. The value of the FM ordering temperature in the RNMO structure is determined by the magnitude of the superexchange spin-transfer integral [32,33]. The low-T_C_ phase of RNMO is attributed to trivalent oxidation states, Ni^3+^ and Mn^3+^, while the high-T_C_ phase has been shown to be characteristic of the atomically ordered Ni^2+^ and Mn^4+^ [34]. It is worth mentioning that other double perovskites, such as La_2_MnCoO_6_, present an important dependence of the magnetic properties on oxygen content in both bulk and thin film forms [35,36]. Oxygen vacancy concentration increases when the films are deposited at low oxygen pressure, which breaks the Ni^2+^–O–Mn^4+^ superexchange ferromagnetic interaction and leads to higher spin disorder, hence decreasing the T_C_.

*M*(*H*) hysteresis loops measured at 10 K with *H* applied in-plane corresponding to two samples grown under oxygen pressure of 350 and 400 mTorr O_2_ with *T*_C_ values about ~200 K are shown in Figure 7c. The shape of the *M*(*H*) loop is practically identical for both samples. Consequently, a similar coercive field (*H*_C_ ≈ 500 Oe) and the same saturation magnetization (*M*_s_ ≈ 5 µ_B_/f.u) were extracted for them. Magnetization remanence *M*_r_ slightly decreases by increasing the oxygen pressure from *M*_r_ ≈ 2.7 µ_B_/f.u to 2.6 µ_B_/f.u, for 350 and 400 mTorr O_2_, respectively. The estimated values of the saturation magnetization equal to the optimal spin-only value of 5 µ_B_/f.u [6,19,22] is encountered in fully ordered Pr_2_NiMnO_6_ systems. Thus, the PNMO films grown at 350 and 400 mTorr O_2_, in spite of the Pr deficiency shown by EPMA (indicating compositions Pr_1.7_Ni_0.9_Mn_1.1_O_6−y_ and Pr_1.5_Ni_1.0_Mn_1.0_O_6−y_, respectively), present very optimal magnetic properties with quite high T_C_ and low temperature saturation magnetization nearby to the maximum expected for Ni^2+^ and Mn^4+^. These magnetic properties are similar to those previously reported by M. P. Singh et al., for PNMO films prepared by PLD [26]. They report good FM properties (i.e., T_C_ ≈ 214 K and M_s_ ≈ 4.35 *µ*_B_/f.u at 10 K) using higher oxygen pressures, i.e., 900 mTorr O_2_ and growth temperatures of 800 °C. Nevertheless, the presence of an additional magnetic transition *T*_C2_ ≈ 126 K was assigned to cation disordering and the coexistence of multiple oxidation states of Ni and Mn. The oxygen pressure used during the deposition of the PNMO thin films plays an important role to stabilize the distinct valence states of Ni and Mn cations in the films, which subsequently affects the FM properties. In our films grown/annealed at 850 °C under different oxygen pressures, the optimal T_C_ and M_S_ (very close to the optimal values [16,21,26]) have been obtained only at oxygen pressures in the 350–400 mTorr O_2_ range.

The analysis of the dependence on the substrate temperature, as mentioned, has been conducted at the optimal oxygen pressure (350 mTorr O_2_). The magnetic properties of the PNMO thin films have been evaluated on films prepared at different growth temperatures (200 °C ≤ T ≤ 900 °C) under a fixed oxygen pressure of 350 mTorr O_2_. In Figure 7d, all M(T) curves are characterized by a single magnetic transition. The evolution of T_C_ as a function of the growth temperature is shown in Figure 7e. Samples grown/annealed at temperatures below or equal to 600 °C show very deficient magnetic properties, with T_C_ ≤ 86 K, and when the growth temperature increases to 800 °C, *T*_C_ rises up to ~210 K. At higher temperatures, from 850 °C on, the ferromagnetic transition temperature slightly decreases again. According to these results, the reduced *T*_C_ in samples grown/annealed below 700 °C could be attributed to the fact that the growth temperature may not be sufficient to promote ordering of Ni and Mn cations. Ordering processes take place by a thermally activated diffusion of cations. In this regard, higher temperatures favor growth kinetics and tend to increase ordering [37,38]. Therefore, growing/annealing at high temperatures (T ≥ 700 °C) promotes Ni/Mn cation ordering and consequently an increase of T_C_ to values very close to that of the bulk one (i.e., T_C_ ≈228 K and Ms ≈ 4.95 µ_B_/f.u at 50 kOe field) [21]. At the same time, the absence of a secondary transition at low temperatures in our films indicates a low concentration of antisite disorder.

*M*(*H*) hysteresis loops corresponding to two PNMO samples with good Curie temperature, grown/annealed at 800 °C (T_C_ ~210 K) and 850 °C (T_C_ ~201 K), measured at 10 K with a magnetic field applied in-plane, are shown in Figure 7f. For both samples, the hysteretic *M*(*H*) behavior can be clearly observed, confirming the ferromagnetic character of the PNMO thin films. The coercive field *H*_C_ was found to be *H*_C_ = 565 Oe and 482 Oe for samples deposited at 800 °C and 850 °C, respectively. At the same time, the remnant magnetization *M*_r_ increases from *M*_r_ ≈ 2.3 µ_B_/f.u to 2.7 µ_B_/f.u and the saturation magnetization increases from *M*_s_ ≈ 4.5 µ_B_/f.u. to 5 µ_B_/f.u, while increasing the growth temperature from 800 °C to 850 °C. On the other hand, in our results, although the *T*_C_ is slightly higher in the sample grown at 800 °C, this film showed a lower saturation magnetization value than that of the film grown at 850 °C. This reduction of the magnetization could be caused by the formation of antiphase boundaries and alternatively some antisite defects (i.e., Ni–O–Ni or Mn–O–Mn) [24,39]. Therefore, films with a T_C_ and Ms very close to the optimal values [16,21,26] have been obtained only at growth temperatures in the 800–850 °C range under an oxygen pressure of 350 mTorr, in spite of their slight Pr deficiency.

The temperature dependence of the resistivity *ρ*(T) measured in the range 150 K ≤ T ≤ 300 K (at zero magnetic field) of a PNMO film with optimized growth conditions (i.e., grown/annealed at 800 °C under 350 mTorr O_2_), displaying good ferromagnetic properties (T_C_ ≈ 210 K and Ms ≈ 4.5 µ_B_/f.u. at 10 K), is shown in Figure 8a. The measurement shows that the resistivity increases with decreasing the temperature, thus exhibiting an insulating behavior in the whole measured temperature range. At lower temperatures (T < 150 K), *ρ*(T) was not measurable due to the high sample resistance. The dependence of *ρ*(T) can be well fitted by using the thermal activation mechanism, described by the Arrhenius law (see Figure 8b) $\rho \left(T\right)=\rho_{0}\exp\left(E_{0}/K_{B}T\right),$ where $\rho_{0}$ is a constant, *E*_0_ is the required activation energy for the conduction process generally at high temperatures, and *K*_B_ is the Boltzmann constant [40,41]. This model can explain the thermally generated charge-transport behavior and activation type mechanism in the insulating state [42]. The fitting of the Arrhenius model was found in a temperature range from 300 K to 195 K. The extracted value of the activation energy *E*_0_ for our PNMO film was found to be *E*_0_ ≈ 0.18 ± 0.1 eV, which is comparable to that of the cation-ordered RNMO compounds [40,42,43].

## 4. Conclusions

In conclusion, we have achieved the epitaxial growth of double perovskite PNMO thin films on (001) STO substrates by the RF magnetron sputtering technique. It is shown that ferromagnetic ordering is significantly influenced by both the oxygen pressure and the growth/annealing temperature. The optimization of the growth conditions has revealed that only under a narrow window of deposition conditions (an oxygen pressure of 350–400 mTorr and growth/annealing temperatures around 800–850 °C), PNMO thin films with good ferromagnetic properties (i.e., T_C_ ≥ 210 K and Ms ≥ 4.5 µ_B_/f.u. at 10K very close to the bulk value) and insulating behavior can be prepared by using AC magnetron sputtering. High annealing temperatures (800–900 °C) promote the appearance of a nanostructured surface, attributed to the formation of NiO_x_ segregates, as previously observed in LNMO systems. Additionally, the XRD patterns showed the existence of a secondary NiO phase. However, although secondary phases are formed (NiO) together with the PNMO phase, films grow epitaxially on the (001) STO substrates. Regarding the stoichiometry of the films (measured by EPMA), it is found that the optimization of the growth parameters does not imply obtaining stoichiometric samples of the expected Pr_2_NiMnO_6_ compound. Particularly, the PNMO films grown/annealed at different growth temperatures under a high oxygen pressure (PO_2_ ≥ 350 mTorr) show a Ni/Mn ratio close to 1 and a Pr:(Ni + Mn) ratio with a certain degree of Pr deficiency, and in spite of this Pr deficiency, the samples exhibit good ferromagnetic and electrical transport properties.

## Figures and Tables

**Figure 1 materials-15-05046-f001:**
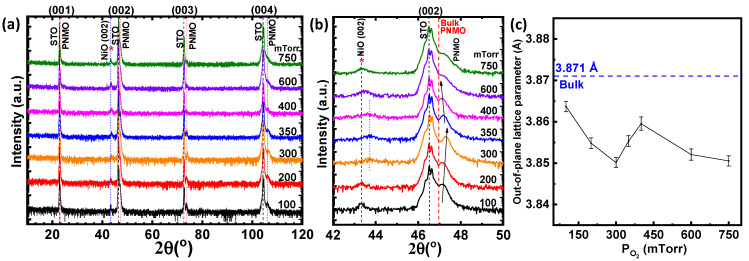
(**a**) XRD θ/2θ scans (in logarithmic scale) of PNMO thin films grown at 850 °C under different oxygen pressures (ranging from 100 to 750 mTorr). The parasitic phase is indicated by (*). (**b**) Zoom of the (002) reflection of both STO and PNMO film and (**c**) out-of-plane *c* lattice parameter as a function of oxygen pressure. The dashed line represents the PNMO bulk value in pseudocubic notation (taken as the cubic root of the unite cell volume per Mn ion, according to the Rietveld refinement of the PNMO target shown in Appendix A).

**Figure 2 materials-15-05046-f002:**
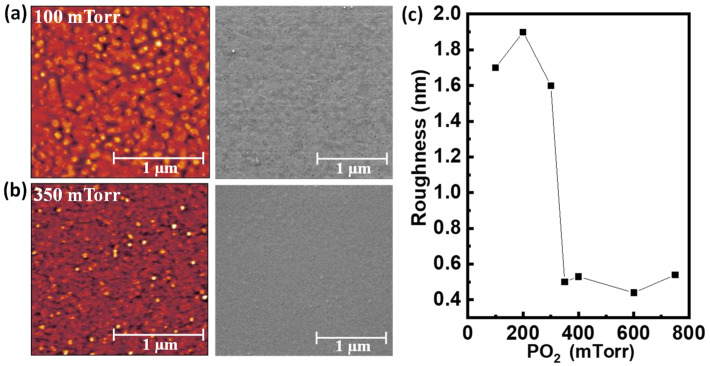
AFM topography (2 × 2 µm^2^ area, left-hand images) and SEM micrographs of corresponding AFM images (right-hand images) of PNMO thin films grown at 850 °C under (**a**) 100 mTorr O_2_ and (**b**) 350 mTorr O_2_. (**c**) Roughness as a function of oxygen pressure of PNMO thin films grown at 850 °C under different oxygen pressures (ranging from 100 to 750 mTorr).

**Figure 3 materials-15-05046-f003:**
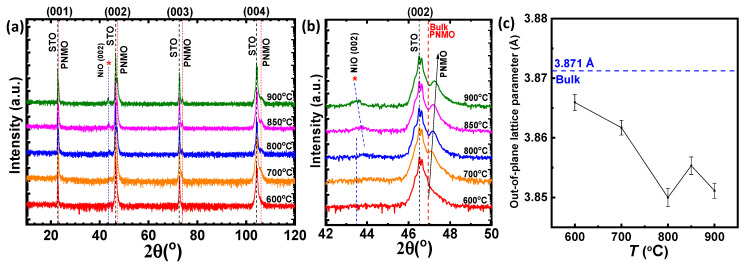
(**a**) XRD θ/2θ scans of PNMO thin films grown at different substrate temperatures (ranging from 600 °C to 900 °C) under 350 mTorr O_2_. The parasitic phase is indicated by (*). (**b**) Zoom of the (002) reflection of both STO and PNMO film and (**c**) out-of-plane *c* lattice parameter as a function of growth temperature. The dashed blue line represents the PNMO bulk-value in pseudocubic notation (taken as the cubic root of the unite cell volume per Mn ion, according to the Rietveld refinement of the PNMO target shown in Appendix A).

**Figure 4 materials-15-05046-f004:**
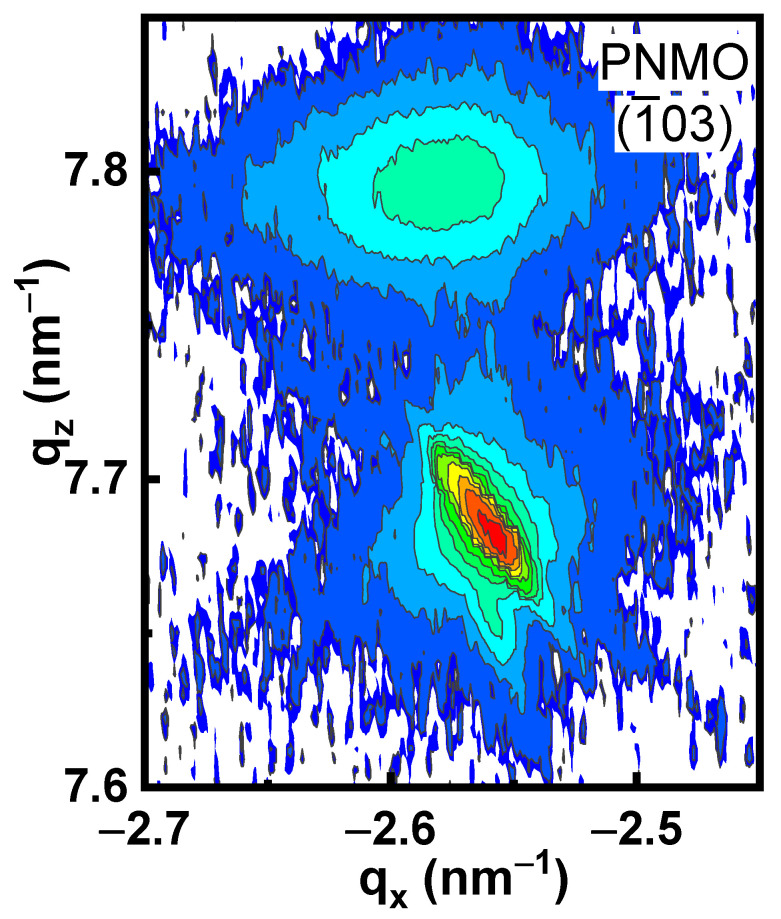
Reciprocal Space Map (RSM) around (−103) reflection of an illustrative PNMO film (thickness, *t* = 47.6 nm) grown/annealed at 800 °C under 350 mTorr O_2_.

**Figure 5 materials-15-05046-f005:**
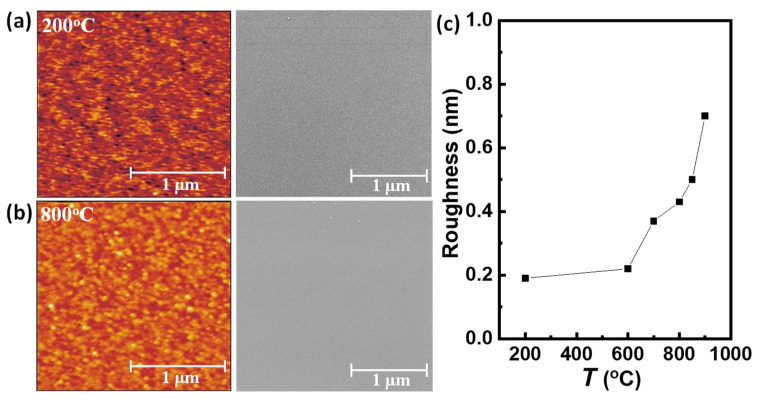
AFM topography (2 × 2 µm^2^ area, left-hand images) and SEM micrographs of corresponding AFM images (right-hand images) of PNMO thin films grown under 350 mTorr O_2_ at growth temperatures of (**a**) 200 °C and (**b**) 800 °C, with in situ annealing at the same growth temperature (1 h under 420 Torr O_2._). (**c**) Roughness as a function of temperature of PNMO thin films grown under 350 mTorr O_2_ at different growth temperatures (ranging from 200 °C to 900 °C).

**Figure 6 materials-15-05046-f006:**
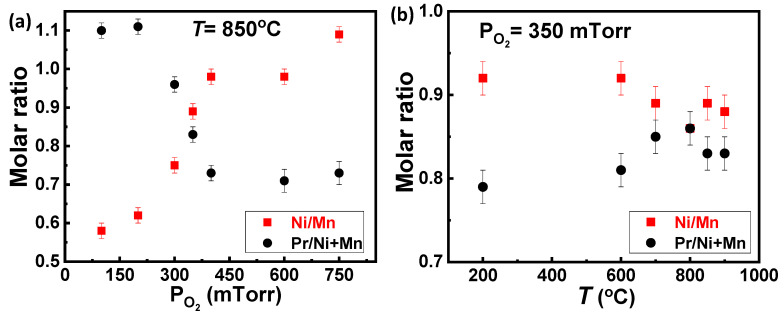
Stoichiometry ratio of PNMO thin films grown on STO (001) substrates (**a**) under different oxygen pressures (ranging from 100 to 750 mTorr) at 850 °C and (**b**) different growth temperatures (200 °C ≤ T ≤ 900 °C) under 350 mTorr O_2_ (obtained by EPMA). Error bars have been estimated from the dispersion of the EPMA results at different point of the samples.

**Figure 7 materials-15-05046-f007:**
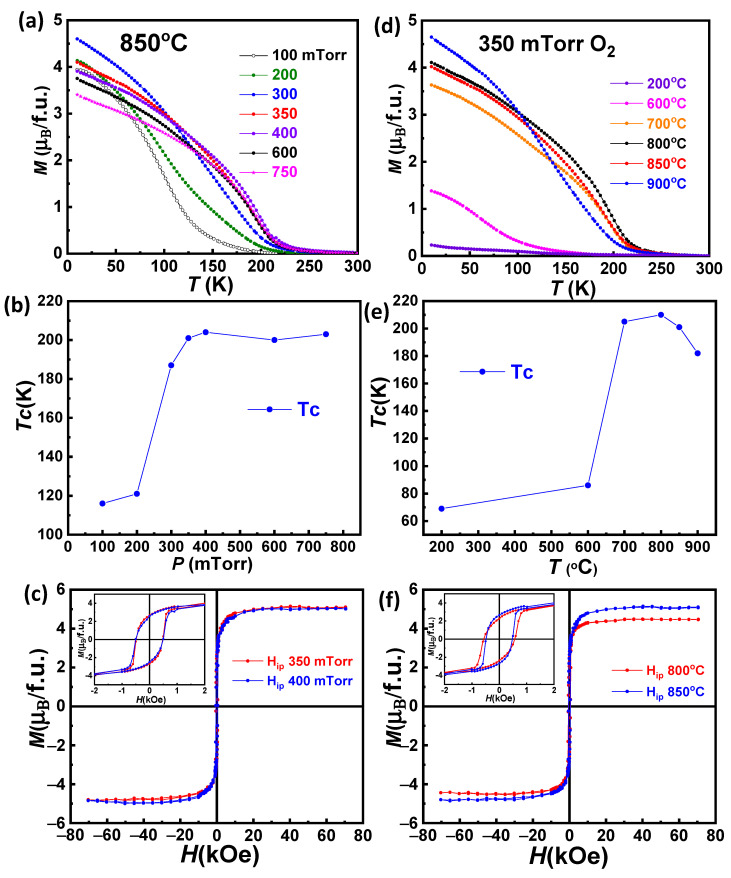
Temperature dependence of in-plane magnetization under an applied field of 5 kOe (µ_0_H = 0.5 T) of PNMO thin films grown: (**a**) at 850 °C under different oxygen pressures, (**d**) at different growth temperatures under 350 mTorr O_2_. Curie temperature *T*_C_ as a function of: (**b**) oxygen pressure and (**e**) growth temperature for the same films. M(H) hysteresis loops of two samples grown/annealed at: (**c**) 850 °C under an oxygen pressure of 350 mTorr and 400 mTorr, (**f**) 800 °C and 850 °C under an oxygen pressure of 350 mTorr, measured at 10 K for H applied in-plane. The inset shows in detail the low field region.

**Figure 8 materials-15-05046-f008:**
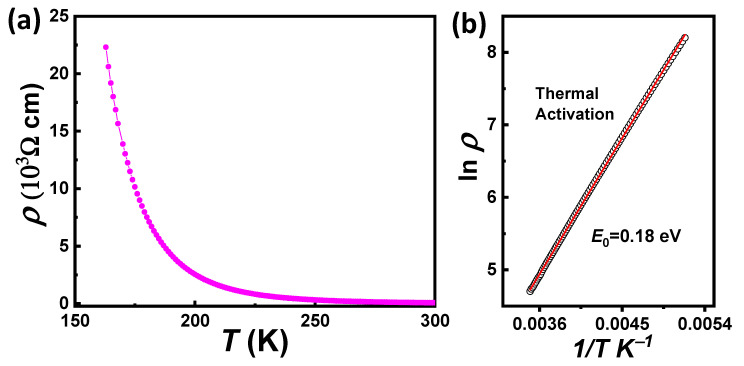
(**a**) Temperature-dependent resistivity of a 47.6 nm-thick PNMO film (grown/annealed at 800 °C under 350 mTorr O_2_) under zero magnetic field. (**b**) Plot of ln *ρ* vs. 1/*T* for the same sample. The red line represents the fitting of the experimental data.

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
