# Peer review of "Optimization of the Growth Process of Double Perovskite Pr2−δNi1−xMn1+xO6−y Epitaxial Thin Films by RF Sputtering"

_materials, 2022, doi:10.3390/ma15145046_

Round 1

Reviewer 1 Report

Bernal-Salamanca et al. presented the results of epitaxial growth of Pr2-δNi1-xMn1+xO6-y thin films on (001)-oriented SrTiO3 substrates by RF magnetron sputtering. The topic of these studies is actual and corresponds to the aims and objectives of the journal. Although the materials presented in the article do not allow to fully evaluate the results obtained. Thus, the manuscript cannot be published in the journal in its current form and should be seriously revised.

1. Many references are missing in the text. The format of references in the text differs from the ones in the reference list.

2. It is necessary to correct subscripts and superscripts throughout the text.

3. Line 36: there are unreadable phrases. Please check all the text and correct language and style.

4. There is no file with supporting information referred to by the authors in the article. This file, as it appears in the text, contains a great amount of information that is needed to understand the research described in the article.

5. Please define the acronym STO when it appears in the text first time.

6. Figures should be inserted into the main text close after to their first citation.

7. How the NiO impurity affects the characteristics of the epitaxial films obtained. How can one get out of formation of the associated phase? Have the authors attempted to do this?

8. For the reader's benefit, it is necessary to briefly compare the results described by the authors in the presented work with those carried out earlier.

The authors have done a lot of experimental work and the article contains interesting conclusions, but needs a thorough revision.

Reviewer 2 Report

attached

Round 2

Reviewer 1 Report

The authors have substantially revised the article, and it may be published in the journal.